# REVISIT SELF-DEBUGGING WITH SELF-GENERATED TESTS FOR CODE GENERATION

## ABSTRACT

Large language models (LLMs) have shown significant advancements in code generation, but still face challenges on tasks beyond their basic capabilities. Recently, the notion of self-debugging has been proposed to boost the performance of code generation by leveraging execution feedback from tests. Despite its promise, the availability of high-quality tests in real-world scenarios is limited. In this context, self-debugging with self-generated tests is a promising solution but lacks a full exploration of its limitations and practical potential. Therefore, we investigate its efficacy on diverse programming problems. To deepen our understanding, we propose two distinct paradigms for the process: post-execution and in-execution self-debugging. Within the scope of self-contained Python programming tasks, we find that post-execution self-debugging struggles on basic problems but shows potential for improvement on competitive ones, due to the bias introduced by self-generated tests. On the other hand, in-execution self-debugging enables LLMs to mitigate the bias by solely leveraging intermediate states during execution, thereby enhancing code generation.

## 1 INTRODUCTION

Large language models (LLMs) have demonstrated considerable progress in code generation, but still face challenges to perform complex programming tasks beyond their basic capabilities. The tasks require LLMs to understand the given natural language specifications and generate programs that could pass all the private tests. Recently, self-debugging has emerged as a promising approach to boost the performance of LLMs in code generation (Chen et al., 2024b; Jiang et al., 2023; Zhong et al., 2024). This approach enables models to refine their own output through an iteration of generation and execution for the programs utilizing *pre-built oracle tests*. However, in real-world scenarios of software development, oracle tests are not available for each code snippet.

To address this challenge, recent studies have introduced *self-generated tests* into self-debugging process (Shinn et al., 2024; Huang et al., 2023; Ridnik et al., 2024). As illustrated in Figure 1, in this framework, the model first generates an initial program and a suite of tests based on the natural language specifications of the problem. The program is then executed on the self-generated tests with an executor (e.g. code interpreter). If it raises any error, the signal or message will be collected as execution feedback, which the model uses to generate a revised version of the program. It helps reduce the reliance on external feedback from humans or stronger models and thus holds the potential to be generally applied in various code generation tasks.

Nonetheless, the efficacy of self-debugging with self-generated tests remains underexplored. Reflexion (Shinn et al., 2024) leverages feedback from self-generated tests to debug but evaluates the code before repair with hidden oracle tests. AlphaCodium (Ridnik et al., 2024) first iterates on public oracle tests and then on model-generated tests with a technique of test anchors. The improvements observed using oracle tests do not accurately demonstrate the true self-debugging capabilities of LLMs. This highlights the need for more transparent evaluation to better understand the inherent debugging potential with self-generated tests. To study this, we first clarify the concept of self-debugging in practice, a scenario wherein the model attempts to debug and repair its own programs without reliance on human supervision or guidance from stronger models. Beyond leveraging the model's intrinsic capabilities, execution feedback from *self-generated tests* also serves as additional signals to help LLMs identify bugs in its programs according to specifications. Depending on the ex-

Figure 1: Overview of self-debugging with execution feedback from self-generated tests. (1) The model generates an initial program along with a suite of tests, based on the specifications of the problem. (2) The program is executed by an executor on the self-generated tests. (3) The feedback from execution is then utilized by the model to produce a revised version of the program.

ecution stage, there are different kinds of information that we can utilize. We propose two paradigms for doing this: post-execution and in-execution self-debugging, as shown in Figure 1. Post-execution self-debugging directly validates correctness by checking whether the output after execution matches the test output or not. In-execution self-debugging allows LLMs to analyze the intermediate runtime states during program execution without knowing the results from post-execution.

**Contributions:** In this paper, we investigate the efficacy of self-debugging with self-generated tests applied to four advanced LLMs: GPT-4o (`2024-05-13`)[1], Claude-3.5-Sonnet[2], Llama-3-70B-Intruct (Dubey et al., 2024) and Qwen2.5-Coder-7B-Instruct (Hui et al., 2024) for self-contained Python programming problems taken from HumanEval (Chen et al., 2021), MBPP (Austin et al., 2021) and LiveCodeBench (Jain et al., 2024). Specifically, we evaluate the models' ability to reflect upon and debug code using information obtained from post-execution and in-execution respectively. We summarize our observations as follows:

- In the context of self-contained Python programming tasks, post-execution self-debugging struggles with relatively basic problems, such as those in HumanEval and MBPP. However, it shows potential for improvement on more challenging programming problems in LiveCodeBench.

- This discrepancy is attributed to the bias introduced by *self-generated tests*, which refers to the misalignment between self-testing labels and true labels for the programs. In addition to the impact of the bias, the efficacy of post-execution self-debugging relies not only on the model's ability to reflect upon feedback but also on the ability to recognize faulty feedback.

- Instead of using unreliable post-execution information, in-execution self-debugging minimizes the bias by solely focusing on the intermediate states during the program execution. The experimental results demonstrate promising improvements for both basic and competitive tasks.

Through our study, we aim to shed light on the practicality of self-debugging with self-generated tests, contributing valuable insights into the future development of LLMs in code generation tasks.

## 2 RELATED WORK

**Code Generation.** Code generation is the automatic production of source code based on natural language descriptions. Large pre-trained language models like the GPT-4 series have shown impressive capabilities in code generation. Researchers have proposed various approaches to enhance the quality of code generated by these models. Some works, like LLaMA series (Touvron et al., 2023a;b; Dubey et al., 2024), focus on optimizing model training, while others aim to improve code quality through post-processing techniques. For example, CodeT (Chen et al., 2022)

---

[1]https://openai.com/index/hello-gpt-4o/
[2]https://www.anthropic.com/news/claude-3-5-sonnet

generates a large number of code and test cases, using the dual agreement to filter the most promising code candidates. Other methods, such as coder-reviewer (Zhang et al., 2023b) and code-ranker (Inala et al., 2022), apply ranking metrics to select optimal code from multiple candidates. Among these post-processing techniques, methods that involve self-debugging have gained considerable attention. Through feedback from execution results, self-debugging allows models to autonomously debug and refine previously generated code, enhancing the final output. Self-debugging does not require increasing the sample budget, making it a cost-effective solution for improving inference efficiency (Zhang et al., 2023a). As a result, self-debugging has been integrated into various LLM-based code generation methods (Yang et al., 2024; Zhang et al., 2024; Dong et al., 2023; Huang et al., 2023). In this work, we revisit these techniques and assess the effectiveness of self-debugging with self-generated tests on both basic and competitive programming benchmarks.

**Self-Debug with LLMs.** As LLMs have evolved, the idea of using models to refine their own output has become more popular. In code generation, several techniques have explored how LLMs can refine the code they generate. Most of these methods rely on prompting LLMs with execution results to improve the code. These methods often rely on pre-existing or generated tests to execute the code, capturing execution information that is then used to refine the output code (Olausson et al. (2024); Wang et al. (2024); Dong et al. (2023); Madaan et al. (2023); Zhang et al. (2023a)). Self-Debugging (Chen et al., 2024b) introduces a framework in which LLMs iteratively debug their own generated code by utilizing execution results and self-generated explanations. Self-Edit (Zhang et al., 2023a) builds on the example tests provided in programming problems for execution to help the model correct its own output. LDB (Zhong et al., 2024) utilizes runtime execution information to help debug generated programs. Jiang et al. (2024) enhance LLM self-debugging by training on an automatically collected dataset for code refinement and explanation. Madaan et al. (2023) conduct a broad evaluation of self-debugging in code models, highlighting that performance can be improved with higher-quality feedback or human intervention. In this work, we aim to explore the potential as well as limitations of execution-based self-debugging methods, particularly with self-generated tests. We provide a detailed analysis of these methods and propose a unified framework in the following Section 3.

## 3 SELF-DEBUGGING WITH SELF-GENERATED TESTS

We focus on evaluating the self-debugging capabilities of large language models (LLMs) through execution on self-generated tests. Figure 1 provides a comprehensive overview of this process. Given a problem with a natural language specification, the LLM (denoted as M) first generates an initial program $C$ along with a suite of test cases, denoted as $\{(X_i, Y_i)\}_{i=1}^N$, where $X_i$ represents the input and $Y_i$ represents the expected output for the $i$-th test. To enhance the model's debugging performance beyond its intrinsic reasoning capabilities, we utilize execution feedback as an additional signal to help the model identify bugs in its generated program according to the problem specification. Specifically, we employ an executor (denoted as E) to run the generated program on the test suite and collect execution information as feedback.

There are various implementations for utilizing execution feedback, which we categorize into two distinct paradigms: **Post-Execution** and **In-Execution** self-debugging. These paradigms reflect the type of information employed in the self-debugging process. Post-execution information refers to content obtained after the program's execution, such as execution outputs or error messages. In contrast, in-execution information refers to intermediate states observed during execution, providing finer-grained insights into the program's behavior. We now formally define these paradigms.

**Post-Execution Self-Debugging.** The paradigm leverages information obtained after the actual execution of the program. A widely adopted implementation involves comparing the execution output with the expected output (Olausson et al., 2024; Wang et al., 2024; Dong et al., 2023; Madaan et al., 2023; Zhang et al., 2023a; Chen et al., 2024b; Jiang et al., 2024), as shown in Figure 1. Consider an initial program $C$ and a generated test set $\{(X_i, Y_i)\}_{i=1}^N$. An executor, denoted as E, processes each input $X_i$, yielding the corresponding execution output $\tilde{Y}_i = \mathrm{E}(C, X_i), i \in [1, N]$. The executor then assesses whether the execution output $\tilde{Y}_i$ aligns with the expected output $Y_i$ to determine if the test is passed. If a discrepancy occurs, the test is marked as failed. The system then utilizes the failed test case $(X_i, Y_i)$, the execution output $\tilde{Y}_i$, and any related error messages to

refine the program. This process encourages the model to generate a revised version of the program, denoted as $\tilde{C} = \mathrm{M}(C, X_i, Y_i, \tilde{Y}_i)$.

**In-Execution Self-Debugging.**   Post-execution self-debugging typically overlooks the intermediate states of the program, which can provide valuable insights for program refinement. To address this limitation, in-execution self-debugging leverages feedback from the intermediate states during program execution (Zhong et al., 2024; Ni et al., 2024; Bouzenia et al., 2023). Formally, a program $C$ can be divided into multiple basic blocks, denoted as $C = [B^1, B^2, ..., B^K]$, where $B^k$ represents the $k$-th basic block and $K$ is the total number of blocks in the execution trace. Each basic block is defined as a linear sequence of program statements with a single entry and a single exit point.

Given a test input $X_i$, $i \in [1, N]$, the executor E initializes the input as the initial variable set $V_i^1$ and executes it through the first block $B^1$. The execution updates the variable set to $V_i^2 = \mathrm{E}(B^1, V_i^1)$, where $V_i^2$ denotes the set of variables after executing block $B^1$. This process is repeated iteratively, with the executor processing $V_i^{k+1} = \mathrm{E}(B^k, V_i^k)$ for each subsequent block $B^k$ until the program execution is complete. The sequence of intermediate states represented as the execution trace $T = [B^1, V_i^1, ..., B^K, V_i^K]$, provides a detailed view of how the program behaves over time. By analyzing this trace, the LLM M identifies potential issues within specific blocks and refines the program accordingly, resulting in the updated version $\tilde{C} = \mathrm{M}(C, X_i, T)$.

## 4 EXPERIMENTS

In this section, we evaluate self-debugging capabilities of advanced LLMs using self-generated tests on self-contained Python programming tasks. We carry out experiments to answer the following research questions: (1) When self-debugging with post-execution information from self-generated tests, what would the performance be like on basic programming problems? (2) Is the performance of post-execution self-debugging consistent across different programming tasks? If not, what is the reason behind it? (3) How does in-execution self-debugging perform when considering the settings above? What is the difference between post-execution and in-execution self-debugging?

### 4.1 EXPERIMENTAL SETUP

**Benchmarks.**   We select three popular code generation benchmarks covering basic and competitive[3] programming problems to comprehensively evaluate the efficacy of self-debugging, including:

- **HumanEval and MBPP**   HumanEval (Chen et al., 2021) consists of 164 programming problems written by humans. Each problem provides a Python function signature and a docstring as its specification. MBPP (Austin et al., 2021) includes 974 programming problems written by contributors through crowdsourcing. Each of these problems features a problem statement, a function signature, and three example tests. To enhance the reliability and accuracy of evaluations, EvalPlus (Liu et al., 2024) extends HumanEval into a more comprehensive version known as HumanEval+ with 80 times more tests than the original HumanEval. Similarly, MBPP+ is an augmentation of the original MBPP, offering 35 times more tests. In our experiments, we use the latest version of MBPP for both base and plus set, which consists of 378 programming problems.

- **LiveCodeBench**   LiveCodeBench (Jain et al., 2024) is a contamination-free benchmark that continuously collects new problems from prominent competitive programming platforms. As of now, LiveCodeBench features a collection of over 600 high-quality programming problems. These problems encompass a wide range of difficulty levels and topics, providing a comprehensive evaluation for the coding capabilities of LLMs. In our experiments, we select 450 problems that were published between September 2023 and September 2024.

**Test Models and Setup.** Generating high-quality tests poses significant challenges as it necessitates a comprehensive understanding of natural language specifications as well as the capabilities of code reasoning (Chen et al., 2024a). Therefore, we investigate the research questions with four advanced chat models: LLaMA-3-70B-Instruct (Dubey et al., 2024) and Qwen2.5-Coder-7B-Instruct

---

[3] In this work, we regard problems in HumanEval, MBPP as basic programming problems, and those in LiveCodeBench as competitive ones according to overall complexity and difficulty.

Table 1: Pass rates after post-execution self-debugging with *oracle tests* on HumanEval and MBPP. The values highlighted in green are increases relative to the initial generation (one-pass).

| Model | Method | #Iteration | HumanEval | | MBPP | |
|---|---|---|---|---|---|---|
| | | | Base | Plus | Base | Plus |
| GPT-4o-2024-05-13 | One-pass | 0 | 92.1 | 87.8 | 91.5 | 76.5 |
| | Self-debug w/ label | 1 | $93.3^{+1.2}$ | $89.0^{+1.2}$ | $92.6^{+1.1}$ | $80.2^{+3.7}$ |
| | | 2 | $94.5^{+2.4}$ | $90.2^{+2.4}$ | $93.4^{+1.9}$ | $81.2^{+4.7}$ |
| | Self-debug w/ detail | 1 | $93.9^{+1.8}$ | $90.2^{+2.4}$ | $92.9^{+1.4}$ | $81.5^{+5.0}$ |
| | | 2 | $95.1^{+3.0}$ | $92.1^{+4.3}$ | $92.6^{+1.1}$ | $83.1^{+6.6}$ |
| Claude-3.5-Sonnet | One-pass | 0 | 94.5 | 89.0 | 92.6 | 77.0 |
| | Self-debug w/ label | 1 | $95.1^{+0.6}$ | $92.1^{+3.1}$ | $93.7^{+1.1}$ | $82.5^{+5.5}$ |
| | | 2 | $96.3^{+1.8}$ | $92.7^{+3.7}$ | $93.4^{+0.8}$ | $83.3^{+6.3}$ |
| | Self-debug w/ detail | 1 | $97.0^{+2.5}$ | $92.1^{+3.1}$ | $91.8^{-0.8}$ | $82.0^{+5.0}$ |
| | | 2 | $97.6^{+3.1}$ | $94.5^{+5.5}$ | $94.2^{+1.6}$ | $86.0^{+9.0}$ |
| LLaMA-3-70B-Instruct | One-pass | 0 | 79.9 | 73.8 | 84.4 | 71.2 |
| | Self-debug w/ label | 1 | $81.7^{+1.8}$ | $77.4^{+3.6}$ | $85.7^{+1.3}$ | $74.9^{+3.7}$ |
| | | 2 | $86.0^{+6.1}$ | $81.1^{+7.3}$ | $86.8^{+2.4}$ | $75.9^{+4.7}$ |
| | Self-debug w/ detail | 1 | $84.1^{+4.2}$ | $80.5^{+6.7}$ | $85.4^{+1.0}$ | $76.5^{+5.3}$ |
| | | 2 | $84.8^{+4.9}$ | $81.7^{+7.9}$ | $86.0^{+1.6}$ | $78.6^{+7.4}$ |
| Qwen2.5-Coder-7B-Instruct | One-pass | 0 | 86.0 | 81.7 | 84.7 | 70.6 |
| | Self-debug w/ label | 1 | $86.0^{+0.0}$ | $82.9^{+1.2}$ | $86.8^{+2.1}$ | $73.8^{+3.2}$ |
| | | 2 | $86.0^{+0.0}$ | $82.9^{+1.2}$ | $86.8^{+2.1}$ | $73.8^{+3.2}$ |
| | Self-debug w/ detail | 1 | $86.6^{+0.6}$ | $83.5^{+1.8}$ | $85.4^{+0.7}$ | $73.8^{+3.2}$ |
| | | 2 | $87.2^{+1.2}$ | $84.1^{+2.4}$ | $86.0^{+1.3}$ | $74.3^{+3.7}$ |

(Hui et al., 2024) with publicly accessible weights, API-served GPT-4o-2024-05-13 and Claude-3.5-Sonnet. We employ a greedy decoding strategy (a temperature of zero) across all generation phases of self-debugging. We design prompts for the initial program generation to ensure that no additional information is introduced by subsequent prompts for program repair. This premise is crucial for us to concentrate on investigating the true self-debugging capabilities of LLMs (Huang et al., 2024). To generate a test suite for each problem, we prompt the model to write ten diverse and extensive tests[4] based on its corresponding natural language specification in a zero-shot manner. For a detailed overview of the prompts used, please refer to the Appendix D.

## 4.2 RQ1: POST-EXECUTION SELF-DEBUGGING STRUGGLES ON BASIC PROBLEMS

In this subsection, we examine the performance of self-debugging techniques using *self-generated tests* on basic programming problems and evaluate how it compares to self-debugging with *oracle tests*. Consistent with implementations in most existing literature, we perform self-debugging by utilizing post-execution information. In this process, program correctness is determined by comparing the actual output with the expected output for a given test case. If the generated program successfully passes all tests, the iterative process terminates, and no further self-debugging is conducted.

**Feedback.** To provide a comprehensive assessment, we consider two different types of feedback that can be utilized from post-execution results. The first type is the correct *label*, which indicates whether the model's previous program was correct or not. If the program is incorrect, an instruction for repair will be provided to the model. The second type is the *detail* of the failure, including the test input, expected output, and execution output. In cases where the program raises an exception during execution, the error message is incorporated into the detail in place of the execution output.

**Results.** We conduct experiments on problems from HumanEval and MBPP using self-generated tests and compare the results to those obtained with oracle tests. Table 1 summarizes the pass rates achieved through self-debugging with oracle tests, showcasing significant improvements as

---

[4]Refer to Appendix C for discussions on the effect of the number of generated tests.

Table 2: Pass rates after post-execution self-debugging with *self-generated tests* on HumanEval and MBPP. The values highlighted in red are declines compared to the initial generation (one-pass).

| Model | Method | #Iteration | HumanEval | | MBPP | |
|---|---|---|---|---|---|---|
| | | | Base | Plus | Base | Plus |
| GPT-4o-2024-05-13 | One-pass | 0 | 92.1 | 87.8 | 91.5 | 76.5 |
| | Self-debug w/ label | 1 | $91.5^{-0.6}$ | $87.2^{-0.6}$ | $92.1^{+0.6}$ | $76.7^{+0.2}$ |
| | | 2 | $91.5^{-0.6}$ | $86.6^{-1.2}$ | $92.9^{+1.4}$ | $77.5^{+1.0}$ |
| | Self-debug w/ detail | 1 | $89.0^{-3.1}$ | $84.1^{-3.7}$ | $91.3^{-0.2}$ | $76.2^{-0.3}$ |
| | | 2 | $91.5^{-0.6}$ | $85.4^{-2.4}$ | $92.6^{+1.1}$ | $76.5^{+0.0}$ |
| Claude-3.5-Sonnet | One-pass | 0 | 94.5 | 89.0 | 92.6 | 77.0 |
| | Self-debug w/ label | 1 | $93.9^{-0.6}$ | $88.4^{-0.6}$ | $92.9^{+0.3}$ | $77.8^{+0.8}$ |
| | | 2 | $93.3^{-1.2}$ | $86.6^{-2.4}$ | $91.5^{-1.1}$ | $76.2^{-0.8}$ |
| | Self-debug w/ detail | 1 | $87.2^{-7.3}$ | $81.1^{-7.9}$ | $90.5^{-2.1}$ | $72.8^{-4.2}$ |
| | | 2 | $87.2^{-7.3}$ | $79.3^{-9.7}$ | $92.1^{-0.5}$ | $75.4^{-1.6}$ |
| LLaMA-3-70B-Instruct | One-pass | 0 | 79.9 | 73.8 | 84.4 | 71.2 |
| | Self-debug w/ label | 1 | $74.4^{-5.5}$ | $65.2^{-8.6}$ | $82.5^{-1.9}$ | $68.3^{-2.9}$ |
| | | 2 | $75.6^{-4.3}$ | $69.5^{-4.3}$ | $83.6^{-0.8}$ | $68.3^{-2.9}$ |
| | Self-debug w/ detail | 1 | $74.4^{-5.5}$ | $66.5^{-7.3}$ | $82.3^{-2.1}$ | $64.8^{-6.4}$ |
| | | 2 | $73.8^{-6.1}$ | $67.1^{-6.7}$ | $80.2^{-4.2}$ | $63.8^{-7.4}$ |
| Qwen2.5-Coder-7B-Instruct | One-pass | 0 | 86.0 | 81.7 | 84.7 | 70.6 |
| | Self-debug w/ label | 1 | $82.9^{-3.1}$ | $78.0^{-3.7}$ | $84.9^{+0.2}$ | $69.8^{-0.8}$ |
| | | 2 | $84.1^{-1.9}$ | $79.3^{-2.4}$ | $83.9^{-0.8}$ | $69.8^{-0.8}$ |
| | Self-debug w/ detail | 1 | $84.1^{-1.9}$ | $76.2^{-5.5}$ | $84.7^{+0.0}$ | $68.0^{-2.6}$ |
| | | 2 | $83.5^{-2.5}$ | $75.6^{-6.1}$ | $85.4^{+0.7}$ | $69.0^{-1.6}$ |

Table 3: Accuracies of self-generated tests on HumanEval and MBPP. Test **Input & Output** are evaluated case-by-case; A test **Suite** is deemed valid if all outputs within the suite are correct.

| Model | HumanEval | | | MBPP | | |
|---|---|---|---|---|---|---|
| | Input | Output | Suite | Input | Output | Suite |
| GPT-4o-2024-05-13 | 97.63% | 89.77% | 59.15% | 94.81% | 85.60% | 58.73% |
| Claude-3.5-Sonnet | 97.68% | 89.14% | 56.71% | 95.75% | 87.37% | 58.47% |
| LLaMA-3-70B-Instruct | 94.53% | 84.69% | 49.39% | 90.81% | 82.08% | 51.85% |
| Qwen2.5-Coder-7B-Instruct | 97.19% | 84.85% | 44.50% | 94.35% | 77.33% | 44.44% |

iterations progress. On the other hand, Table 2 presents the results when using self-generated tests. We noted declines across all benchmarks for Llama-3-70b-instruct and Qwen2.5-coder-7b-instruct. For other models, it shows a consistent decrease on HumanEval. The performance on MBPP may improve initially, but with more detailed feedback and iterations, it will ultimately become worse than the initial generation.

**Analysis on generated tests.** To better understand the reliability of tests generated by the model itself, we employ program contracts and canonical solutions provided by the benchmarks to evaluate the validity of test inputs and outputs respectively. Program contracts consist of assertions that specify conditions necessary for a valid input. We place these contracts at the beginning of the function and pass the generated test input to it. Please refer to Appendix B for detailed implementation. If there is no assertion error, the test input is considered valid. For test output validation, we collect the actual execution output using canonical solutions, given a valid input, to confirm if the output aligns with the expected output. Furthermore, we calculate the overall accuracy for the entire test suite. A test suite is deemed valid if all generated test outputs are correct for a given problem.

Table 3 summarizes the results. GPT-4o and Claude-3.5-sonnet demonstrate superior capability in producing high-quality tests compared to others, yet they remain prone to generating unreliable tests

Table 4: Pass rates after post-execution self-debugging with *self-generated tests* on LiveCodeBench.

| Model | Method | #Iteration | Easy | Medium | Hard | Overall |
|---|---|---|---|---|---|---|
| GPT-4o-2024-05-13 | One-pass | 0 | 89.3 | 33.1 | 6.0 | 46.0 |
| | Self-debug w/ label | 1 | $89.9^{+0.6}$ | $41.1^{+8.0}$ | $6.0^{+0.0}$ | $49.3^{+3.3}$ |
| | | 2 | $89.9^{+0.6}$ | $40.0^{+6.9}$ | $6.9^{+0.9}$ | $49.1^{+3.1}$ |
| | Self-debug w/ detail | 1 | $85.5^{-3.8}$ | $36.0^{+2.9}$ | $8.6^{+2.6}$ | $46.4^{+0.4}$ |
| | | 2 | $87.4^{-1.9}$ | $38.3^{+5.2}$ | $8.6^{+2.6}$ | $48.0^{+2.0}$ |
| Claude-3.5-Sonnet | One-pass | 0 | 93.1 | 48.0 | 16.4 | 55.8 |
| | Self-debug w/ label | 1 | $89.9^{-3.2}$ | $49.1^{+1.1}$ | $17.2^{+0.8}$ | $55.3^{-0.5}$ |
| | | 2 | $91.2^{-1.9}$ | $49.7^{+1.7}$ | $16.4^{+0.0}$ | $55.8^{+0.0}$ |
| | Self-debug w/ detail | 1 | $89.9^{-3.2}$ | $49.1^{+1.1}$ | $13.8^{-2.6}$ | $54.4^{-1.2}$ |
| | | 2 | $85.5^{-7.6}$ | $43.3^{-4.7}$ | $8.6^{-7.8}$ | $49.3^{-6.5}$ |
| LLaMA-3-70B-Instruct | One-pass | 0 | 72.3 | 10.3 | 2.6 | 30.2 |
| | Self-debug w/ label | 1 | $66.0^{-6.3}$ | $9.1^{-1.2}$ | $3.4^{+0.8}$ | $27.8^{-2.4}$ |
| | | 2 | $64.8^{-7.5}$ | $10.9^{+0.6}$ | $2.6^{+0.0}$ | $27.8^{-2.4}$ |
| | Self-debug w/ detail | 1 | $56.6^{-15.7}$ | $10.9^{+0.6}$ | $4.3^{+1.7}$ | $25.3^{-4.9}$ |
| | | 2 | $63.5^{-8.8}$ | $12.0^{+1.7}$ | $2.6^{+0.0}$ | $27.8^{-2.4}$ |
| Qwen2.5-Coder-7B-Instruct | One-pass | 0 | 74.8 | 23.4 | 8.6 | 35.8 |
| | Self-debug w/ label | 1 | $69.8^{-5.0}$ | $24.0^{+0.6}$ | $8.6^{+0.0}$ | $34.2^{-1.6}$ |
| | | 2 | $71.7^{-3.1}$ | $23.4^{+0.0}$ | $8.6^{+0.0}$ | $34.7^{-1.1}$ |
| | Self-debug w/ detail | 1 | $69.2^{-5.6}$ | $20.0^{-3.4}$ | $8.6^{+0.0}$ | $32.4^{-3.4}$ |
| | | 2 | $66.7^{-8.1}$ | $21.1^{-2.3}$ | $8.6^{+0.0}$ | $32.0^{-3.8}$ |

based on natural language specifications. For all the models, predicting test outputs proves to be a more challenging task than generating test inputs.

In post-execution settings, incorrect test outputs introduce ambiguity into the self-debugging process. We present an example on HumanEval with GPT-4o in Figure 3 in Appendix A. When a test fails, the model is expected to determine whether the failure is due to bugs in the program or errors in the test. This uncertainty complicates the self-debugging process and necessitates a further investigation into the effects of testing on self-generated tests, as discussed in the following Section 4.3. Our experiments reveal that *post-execution self-debugging struggles with basic programming tasks like HumanEval and MBPP*. While post-execution information with self-generated tests is leveraged, self-debugging remains a bottleneck, limiting improvements beyond initial generation.

### 4.3 RQ2: Bias from Self-Testing Leads to Inconsistency across Tasks

To comprehensively evaluate the performance of self-debugging on diverse programming tasks, we conducted post-execution self-debugging experiments using problems from LiveCodeBench. The problems in LiveCodeBench are classified into three distinct difficulty levels: easy, medium, and hard. We report the pass rate achieved at each level of difficulty, as well as the overall performance.

**Results.** Table 4 summarizes the results of post-execution self-debugging with self-generated tests on LiveCodeBench. *We observed that for GPT-4o, self-debugging using label feedback leads to improvements across problems of all difficulty levels. This is notably in contrast to the performance on HumanEval and MBPP.* However, when detailed feedback is provided, there is a decline in performance on easy problems. For other models including Claude-3.5-Sonnet, the overall performance decreases due to significant declines on easy problems. Moreover, despite incorporating more post-execution information, the overall performance with detailed feedback remains inferior to that achieved with label feedback.

**Analysis.** To investigate the reasons behind the inconsistent results on basic and competitive programming problems, we delve into the impact on testing programs with self-generated tests. We acknowledge that the models even advanced LLMs are likely to generate inaccurate tests. Therefore, a program that is actually correct might fail some of the generated tests, resulting in a false negative (FN) label. On the other hand, a flawed program might pass all the test cases, leading to

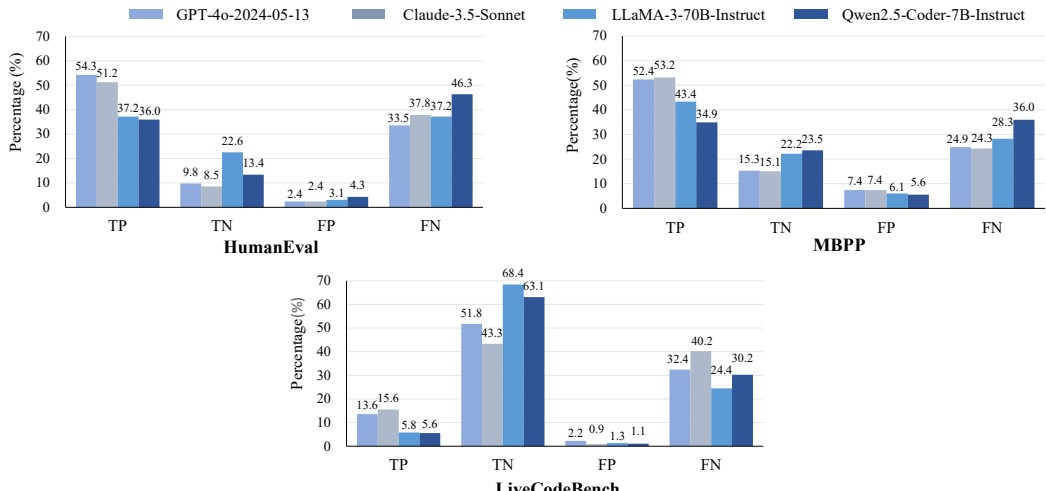

Figure 2: The label changes when evaluating the programs with self-generated tests on HumanEval, MBPP and LiveCodeBench. True Positive (TP): correct programs pass tests; True Negative (TN): incorrect programs fail tests; False Positive (FP): incorrect programs pass tests; False Negative (FN): correct programs fail tests.

a false positive (FP) label. This could prevent necessary updates and prematurely present a buggy program. The misalignment between self-testing labels and true labels highlights the bias introduced by self-generated tests for program evaluation.

We present an analysis of label changes with generated tests after the first iteration of self-debugging, as illustrated in Figure 2. Given the implementation of self-debugging, only programs identified with negative labels during the iteration would perform further repair. Therefore, our focus is primarily on the distribution of different negative labels. We observed that testing on self-generated tests is more likely to result in false negative labels than true negative ones on both HumanEval and MBPP. However, a different pattern emerges on LiveCodeBench, where false negatives are more than true negatives. *This discrepancy is primarily due to lower performance on more challenging programming tasks, where negative labels from self-testing are more likely to align with the actual labels of the generated programs.* Relying solely on labels during self-debugging inadvertently reduces the bias introduced by the self-generated tests, thereby increasing the prevalence of true negative labels. However, when incorrect details are included in feedback, the performance declines compared to using only the label for self-debugging.

Generating high-quality tests from natural language specifications continues to present a substantial challenge in the field. When self-testing results in a false negative due to invalid tests, it is crucial for the model to accurately identify the errors within the feedback and keep the original programs intact. The efficacy of post-execution self-debugging, depends not only on the model's ability to identify the defects in its own programs when presented with true negative labels but also on its ability to recognize the faulty execution feedback given false negatives.

## 4.4 RQ3: IN-EXECUTION REASONING HELPS SELF-DEBUGGING

In this subsection, we examine the efficacy of in-execution self-debugging across programming benchmarks. Drawing inspiration from the implementation presented in LDB (Zhong et al., 2024), we divide a program into basic blocks based on nodes in its control flow graph (CFG). Then we collect the intermediate runtime states before and after these basic blocks during program execution to facilitate in-execution self-debugging. However, the labels (whether the program is correct or not) and details of the execution results, which we regard as post-execution information illustrated in Section 4.2, are not accessible for the models. Therefore, the models must determine program correctness merely based on the test input and corresponding intermediate states, analyzing each block individually.

Table 5: Pass rates after in-execution self-debugging with *self-generated tests* on HumanEval and MBPP.

| Model | Method | #Iteration | HumanEval | | MBPP | |
|---|---|---|---|---|---|---|
| | | | Base | Plus | Base | Plus |
| GPT-4o-2024-05-13 | One-pass | 0 | 92.1 | 87.8 | 91.5 | 76.5 |
| | Self-debug w/ trace | 1 | $93.3^{+1.2}$ | $89.0^{+1.2}$ | $92.1^{+0.6}$ | $77.8^{+1.3}$ |
| | | 2 | $93.3^{+1.2}$ | $88.4^{+0.6}$ | $92.9^{+1.4}$ | $79.1^{+2.6}$ |
| Claude-3.5-Sonnet | One-pass | 0 | 94.5 | 89.0 | 92.6 | 77.0 |
| | Self-debug w/ trace | 1 | $93.9^{-0.6}$ | $89.6^{+0.6}$ | $95.0^{+2.4}$ | $77.2^{+0.2}$ |
| | | 2 | $93.9^{-0.6}$ | $87.2^{-1.8}$ | $93.9^{+1.3}$ | $76.2^{-0.8}$ |
| LLaMA-3-70B-Instruct | One-pass | 0 | 79.9 | 73.8 | 84.4 | 71.2 |
| | Self-debug w/ trace | 1 | $81.1^{+1.2}$ | $70.1^{-3.7}$ | $84.7^{+0.3}$ | $69.6^{-1.6}$ |
| | | 2 | $83.5^{+3.6}$ | $74.4^{+0.6}$ | $84.4^{+0.0}$ | $69.6^{-1.6}$ |
| Qwen2.5-Coder-7B-Instruct | One-pass | 0 | 86.0 | 81.7 | 84.7 | 70.6 |
| | Self-debug w/ trace | 1 | $86.6^{+0.6}$ | $82.3^{+0.6}$ | $84.9^{+0.2}$ | $71.4^{+0.8}$ |
| | | 2 | $86.6^{+0.6}$ | $82.3^{+0.6}$ | $85.2^{+0.5}$ | $72.0^{+1.4}$ |

Table 6: Pass rates after in-execution self-debugging with *self-generated tests* on LiveCodeBench.

| Model | Method | #Iteration | Easy | Medium | Hard | Overall |
|---|---|---|---|---|---|---|
| GPT-4o-2024-05-13 | One-pass | 0 | 89.3 | 33.1 | 6.0 | 46.0 |
| | Self-debug w/ trace | 1 | $91.2^{+1.9}$ | $34.9^{+1.8}$ | $6.0^{+0.0}$ | $47.3^{+1.3}$ |
| | | 2 | $91.8^{+2.5}$ | $34.9^{+1.8}$ | $6.0^{+0.0}$ | $47.6^{+1.6}$ |
| Claude-3.5-Sonnet | One-pass | 0 | 93.1 | 48.0 | 16.4 | 55.8 |
| | Self-debug w/ trace | 1 | $95.0^{+1.9}$ | $49.1^{+1.1}$ | $17.2^{+0.8}$ | $57.1^{+1.3}$ |
| | | 2 | $93.7^{+0.6}$ | $48.6^{+0.6}$ | $17.2^{+0.8}$ | $56.4^{+0.6}$ |
| LLaMA-3-70B-Instruct | One-pass | 0 | 72.3 | 10.3 | 2.6 | 30.2 |
| | Self-debug w/ trace | 1 | $73.0^{+0.7}$ | $11.4^{+1.1}$ | $3.4^{+0.8}$ | $31.1^{+0.9}$ |
| | | 2 | $71.1^{-1.2}$ | $12.0^{+1.7}$ | $3.4^{+0.8}$ | $30.7^{+0.5}$ |
| Qwen2.5-Coder-7B-Instruct | One-pass | 0 | 74.8 | 23.4 | 8.6 | 35.8 |
| | Self-debug w/ trace | 1 | $75.5^{+0.7}$ | $24.0^{+0.6}$ | $8.6^{+0.0}$ | $36.2^{+0.4}$ |
| | | 2 | $76.1^{+1.3}$ | $24.0^{+0.6}$ | $8.6^{+0.0}$ | $36.4^{+0.6}$ |

**Results.** The results of in-execution self-debugging on HumanEval and MBPP are detailed in Table 5. We observe that self-debugging gains notable improvement for GPT-4o and Qwen2.5-coder-7b-instruct when utilizing in-execution information. Specifically, GPT-4o's pass rate increases continuously from 76.5% to 79.1% after two iterations of self-debugging on MBPP. For Claude-3.5-Sonnet, performance improves after the first iteration on both benchmarks and for Llama-3-70b-instruct, the pass rate surpasses the baseline on HumanEval-plus after the second iteration. However, there is a slight degradation in performance in certain tasks and iterations compared to the initial generation. Furthermore, Table 6 summarizes the results on LiveCodeBench, which shows the effectiveness of the in-execution self-debugging for all the models on competitive problems.

**Analysis.** Experimental results indicate that *in-execution self-debug is a potentially effective way by leveraging runtime execution information on both basic and competitive programming problems.* It segments a program into basic blocks and allows LLMs to delve into the precise intermediate states during the execution process. The intermediate states serve as additional cues for program repair and enhancement, significantly mitigating the bias introduced by self-generated tests. Nonetheless, self-debugging with in-execution information depends heavily on the LLMs' code reasoning capabilities and lacks formal guarantees of success, as the pass rate drops for Llama-3-70b-instruct on MBPP. We expect that improvements in LLM capabilities will enhance the efficacy of this paradigm.

To conclude, post-execution self-debugging utilizes final execution results to reflect upon and debug programs. However, the unreliability of the self-generated tests could bias the model away from the

correct answer. Although this can provide some relief on challenging tasks, it is not a long-term solution, especially when those competitive programming problems can also be solved well over time. On the contrary, in-execution self-debugging allows the models to perform fine-grained feedback solely on the intermediate states during the execution process, without knowing the information from biased self-testing. It shows the potential to better align the programs with the requirements in real-world scenarios. Please refer to Appendix A for a detailed comparison of these two paradigms.

## 5    DISCUSSION

**Directions for future work.**    In this work, we demonstrate that post-execution self-debugging with self-generated tests struggles on basic problems due to biased evaluations, despite the significant potential shown by LLMs in automated test generation. This highlights the necessity for the research community to focus on the quality of LLM-generated tests before utilizing execution feedback derived from them. Developing techniques that enhance high-quality test synthesis is crucial to mitigate bias for post-execution self-debugging. It could be beneficial to implement an iterative refinement process wherein execution information is leveraged to improve the tests. This could involve using techniques like test-driven development where tests are continuously updated based on code changes and debugging outcomes.

As demonstrated in Section 4.4, leveraging enriched runtime information from execution is a promising avenue for self-debugging. In particular, in-execution self-debugging has shown superior performance compared to post-execution in certain tasks, suggesting that more nuanced and reliable feedback leads to better performance. Designing more sophisticated methods for collecting and analyzing runtime information is a promising direction for further enhancing self-debugging capabilities. For instance, improving the intelligibility of execution trace representations for LLMs may prove beneficial (Ni et al., 2024). Additionally, beyond variables, other types of runtime information, such as code coverage and execution paths, may also be utilized effectively (Chen et al., 2024a).

Effective self-debugging with self-generated tests hinges on several core capabilities of LLMs. In terms of refinement, the model should be capable of accurately recognizing and localizing faults within the program. Additionally, more advanced reasoning capabilities are needed to analyze execution feedback thoroughly. The model should comprehend the relationship between the code logic and the feedback, thereby deducing the runtime structure of program statements and variables.

**Applications.**    Self-debugging opens up possibilities for developing more advanced LLMs without reliance on human supervision or guidance from stronger models (Burns et al., 2023). Traditionally, human-generated test cases serve as a strong supervisory signal for aligning code generation, but the collection of such tests is labor-intensive, leading to a sparsity of labeled data for effective code refinement. Self-generated tests, by contrast, offer a viable path for self-improvement (Tao et al., 2024). They alleviate the burden of manual test generation and pave the way toward truly autonomous self-correcting code generation systems (Chen et al., 2024b).

## 6    CONCLUSION

This paper investigates the concept of self-debugging in code generation for LLMs, with a focus on leveraging self-generated tests. We establish a structured framework for self-debugging which is essential for real-world applications where high-quality annotations and human supervision are often unavailable. We introduce and formalize two distinct paradigms within the execution-then-feedback process: post-execution and in-execution self-debugging. Through comprehensive experiments on both basic and competitive programming tasks, our findings highlight the unique strengths and weaknesses. Specifically, we observe that: 1) post-execution self-debugging encounters difficulties in basic tasks; 2) bias from self-generated tests can lead to inconsistency across different levels of problems; and 3) in-execution self-debugging, which leverages intermediate runtime information, consistently outperforms post-execution approach on both basic and competitive tasks, indicating significant potential for future development. Overall, our work provides valuable insights into the mechanics of self-debugging using self-generated tests, paving the way toward more autonomous and self-evolving code generation systems.

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

# A  CASE STUDY

In our experiments, we observe that in-execution self-debugging, which leverages runtime information, consistently outperforms post-execution one across various levels of self-contained programming tasks. To better understand the unique strengths and weaknesses of these two paradigms, we provide an example involving GPT-4o in Figure 3. It illustrates different outcomes of post-execution self-debugging with detailed test feedback and in-execution self-debugging with execution traces.

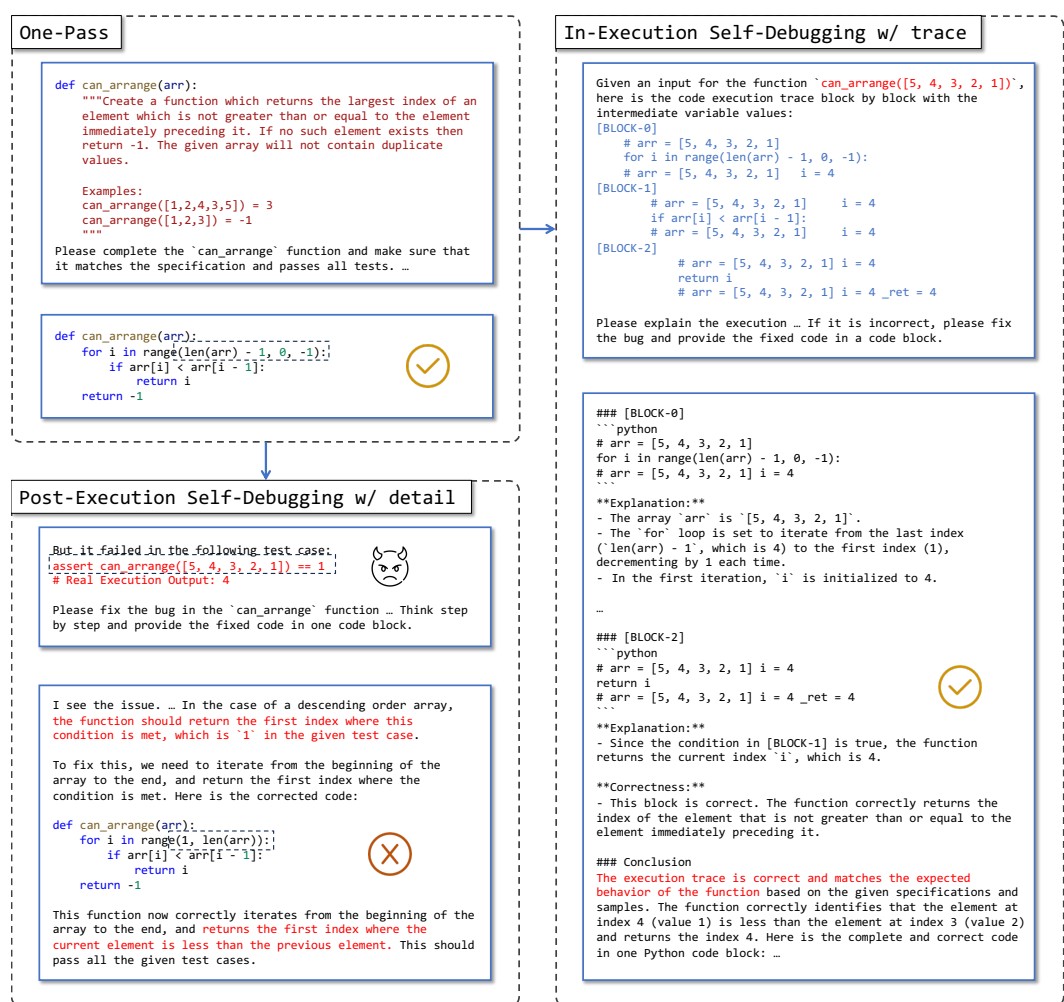

Figure 3: An example with GPT-4o performing both post and in-execution self-debugging on a problem from HumanEval (`HumanEval/135`) respectively. Post-execution self-debugging wrongly corrects the program while in-execution self-debugging manages to keep the original answer.

In this example, the completion for the `can_arrange` function is initially correct. However, it is evaluated against an erroneous self-generated test that, according to the specification, should return `4` instead of `1`. This discrepancy makes the model alter its original correct interpretation of the condition in the problem, thereby leading to a wrongly revised program. Feedback from post-execution on erroneous self-generated tests biases the model away from the specification of the problem. By contrast, in-execution self-debugging leverages test inputs and their corresponding runtime information to assess program correctness. As depicted in Figure 3, this approach enables the model to perform a fine-grained analysis on the execution trace block by block without access to the potential biases introduced by self-generated tests. The model eventually confirms that the trace aligns with the expected behavior of the function.

## B  EXAMPLES OF PROGRAM CONTRACTS

```python
# HumanEval/21

def rescale_to_unit(numbers: List[float]) -> List[float]:
    """ Given list of numbers (of at least two elements), apply a linear transform to that list,
    such that the smallest number will become 0 and the largest will become 1
    >>> rescale_to_unit([1.0, 2.0, 3.0, 4.0, 5.0])
    [0.0, 0.25, 0.5, 0.75, 1.0]
    """

    assert all(type(x) in [int, float] for x in numbers), "invalid inputs" # $_CONTRACT_$
    assert len(numbers) >= 2, "invalid inputs" # $_CONTRACT_$
    assert max(numbers) > min(numbers), "invalid inputs" # $_CONTRACT_$

    ma, mi = max(numbers), min(numbers)
    k = 1 / (ma - mi)
    return list(map(lambda x: (x - mi) * k, numbers))
```

```python
# MBPP/439

"""
Write a function to join a list of multiple integers into a single integer.
assert multiple_to_single([11, 33, 50])==113350
"""

def multiple_to_single(L):

    assert isinstance(L, list), "invalid inputs" # $_CONTRACT_$
    assert len(L) > 0, "invalid inputs" # $_CONTRACT_$
    assert all(isinstance(item, int) for item in L), "invalid inputs" # $_CONTRACT_$
    assert all(item > 0 for item in L[1:]), "invalid inputs" # $_CONTRACT_$

    return int(''.join(map(str,L)))
```

Figure 4: Examples of program contracts in HumanEval and MBPP. Program contracts consist of assertions that specify conditions necessary for a valid input.

## C  ANALYSIS ON THE NUMBER OF SELF-GENERATED TESTS

To investigate the effect of the number of self-generated tests, we employ GPT-4o-2024-05-13 to generate increasing numbers of tests $N = [10, 15, 20]$ for each programming problem in HumanEval and MBPP. The accuracies of these generated tests are summarized in Table 7.

Table 7: Accuracies of increasing sizes of self-generated test suites on HumanEval and MBPP.

| #Num of Tests | HumanEval | | | MBPP | | |
|---|---|---|---|---|---|---|
| | Input | Output | Suite | Input | Output | Suite |
| 10 | 97.63% | 89.77% | 59.15% | 94.81% | 85.60% | 58.73% |
| 15 | 97.89% | 88.86% | 52.44% | 94.96% | 85.27% | 53.70% |
| 20 | 98.11% | 86.01% | 48.17% | 95.10% | 82.94% | 50.53% |

As the number of self-generated tests increases, the presence of more challenging edge cases also rises, consequently reducing the accuracy of the test suites. Specifically, when the model generates up to 20 tests per problem, the accuracy of the test suite decreases from 59.15% to 48.17% for HumanEval and from 58.73% to 50.53% for MBPP. We further evaluate the performance of both post-execution self-debugging with detailed feedback and in-execution self-debugging.

Table 8: Pass rates after post-execution self-debugging with detailed feedback and in-execution self-debugging on HumanEval and MBPP when using different sizes of the self-generated test suite. The values highlighted in red or green are changes relative to the initial generation (one-pass).

| Method | #Num of Tests | HumanEval | | MBPP | |
|---|---|---|---|---|---|
| | | Base | Plus | Base | Plus |
| One-pass | 0 | 92.1 | 87.8 | 91.5 | 76.5 |
| Self-debug w/ detail | 10 | $89.0^{-3.1}$ | $84.1^{-3.7}$ | $91.3^{-0.2}$ | $76.2^{-0.3}$ |
| | 15 | $88.4^{-3.7}$ | $84.1^{-3.7}$ | $91.3^{-0.2}$ | $75.9^{-0.6}$ |
| | 20 | $87.8^{-4.3}$ | $83.5^{-4.3}$ | $90.7^{-0.8}$ | $75.9^{-0.6}$ |
| Self-debug w/ trace | 10 | $93.3^{+1.2}$ | $89.0^{+1.2}$ | $92.1^{+0.6}$ | $77.8^{+1.3}$ |
| | 15 | $92.7^{+0.6}$ | $88.4^{+0.6}$ | $92.1^{+0.6}$ | $78.0^{+1.5}$ |
| | 20 | $93.3^{+1.2}$ | $88.4^{+0.6}$ | $91.8^{+0.3}$ | $77.2^{+0.7}$ |

The results in Table 8 indicate that with an increased number of self-generated tests, the performance of post-execution self-debugging experiences a slight decline on both HumanEval and MBPP. It is attributed to the lower test accuracy leading to a higher rate of false negatives, thereby hindering the efficacy of post-execution self-debugging. Conversely, in-execution self-debugging leveraging intermediate runtime traces shows a consistent improvement over the initial generation.

# D PROMPTS

```
Here is the given code to do completion:
```python
{prompt}
```

Please complete the `{entry_point}` function and make sure that it matches the
specification and passes all tests. You are not allowed to modify the given function
signature. Think step by step and provide all completed codes in one code block.
```

Figure 5: Code generation prompt for HumanEval.

```
Here is the given problem to solve:
```python
{prompt}
```

Please implement the `{entry_point}` function and make sure that it matches the
specification and passes all tests. You are not allowed to modify the given function
name and arguments in the test examples. Think step by step and provide all completed
codes in one code block.
```

Figure 6: Code generation prompt for MBPP.

```
Here is the given programming problem to solve:
{content}

Please generate a correct python program that matches the specification and passes all
tests. Think step by step. You will use the following starter code to write the
solution to the problem and enclose your code within delimiters.
```python
{starter_code}
```
```

Figure 7: Code generation prompt for functional-input question in LiveCodeBench.

```
Here is the given programming problem to solve:
{content}

Please generate a correct python program that matches the specification and passes all
tests. Read the inputs from stdin solve the problem and write the answer to stdout (do
not directly test on the sample inputs). Think step by step and enclose your code
within delimiters as follows:
```python
# YOUR CODE HERE
```
```

Figure 8: Code generation prompt for stdin-input question in LiveCodeBench.

```
Here is the given code to do completion:
```python
{prompt}
```
Please provide ten comprehensive and valid test cases to verify whether the
`{entry_point}` function correctly solves the problem. You are not allowed to
implement the function. Think step by step and provide all test cases in one code
block.

The format of test cases should be:
```python
assert {entry_point}(input) == expected_output, "Test Case Description"
```
```

Figure 9: Test generation prompt for HumanEval.

```
Here is the given problem to solve:
```python
{prompt}
```
Please provide ten comprehensive and valid test cases to verify whether the
`{entry_point}` function correctly solves the problem. You are not allowed to
implement the function. Think step by step and provide all test cases in one code
block.

The format of test cases should be:
```python
assert {entry_point}(input) == expected_output, "Test Case Description"
```
```

Figure 10: Test generation prompt for MBPP.

```
Here is the given programming problem to solve:
{content}

Please provide ten comprehensive test samples based on the specification and follow
the format of the given sample.

Your response should be organized like below and no extra information is allowed
(including explanation):
[Input]
<your input here>
[Output]
<your output here>
[Input]
...
```

Figure 11: Test generation prompt for LiveCodeBench.

```
{error}Please fix the bug in the `{entry_point}` function and make sure that the fixed
code matches the specification and passes all tests. You are not allowed to modify the
given function signature. Think step by step and provide the fixed code in one code
block.
```

Figure 12: Debugging prompt for HumanEval.

```
{error}Please fix the bug in the `{entry_point}` function and make sure that the fixed
code matches the specification and passes all tests. You are not allowed to modify the
given function name and arguments in the test examples. Think step by step and provide
the fixed code in one code block.
```

Figure 13: Debugging prompt for MBPP.

```
{error}Please fix the bug in the code and make sure that the fixed code matches the
specification and passes all tests.
You will use the following starter code to write the solution to the problem and
enclose your code within delimiters.
```python
{starter_code}
```
```

Figure 14: Debugging prompt for functional-input question in LiveCodeBench.

```
{error}Please fix the bug in the code and make sure that the fixed code matches the
specification and passes all tests.
Read the inputs from stdin solve the problem and write the answer to stdout (do not
directly test on the sample inputs). Enclose your code within delimiters as follows.
```python
# YOUR CODE HERE
```
```

Figure 15: Debugging prompt for stdin-input question in LiveCodeBench.

```
Given an input for the function `{test}`, here is the code execution trace block by
block with the intermediate variable values as reference:
{trace}

Please explain the execution FOR EACH BLOCK and answer whether this program is correct
or not based on the specifications and given samples in the problem. If the program is
correct, please restate it in one python code block. If it is incorrect, please fix
the bug and provide the fixed code in a code block.
```

Figure 16: Prompt for in-execution self-debugging.

