# OpenReview forum: "Revisit Self-Debugging with Self-Generated Tests for Code Generation"
_ICLR.cc/2025/Conference — Submitted to ICLR 2025_

### Official Review · Reviewer_edps · 2024-10-26

**Soundness:** 4
**Presentation:** 3
**Contribution:** 3
**Rating:** 8
**Confidence:** 4

**Summary:**

This paper looks at "self-debugging", i.e. when the LLM is tasked with debugging and repairing programs it has generated that do not pass the tests.
This is a problem that has received much attention in the literature in the last year or two; the key novelty here is:
- Looking at the impact of having the tests themselves be synthesized by the model (which risks leading it astray during debugging, if the tests are spurious)
- The second set of experiments, when the authors evaluate what they call "in-execution" repair. In the usual setting (which the authors call "post-execution"), the LLM is simply told which test failed (and, possibly, the manner in which it failed); in the "in-execution" setting, the LLM is instead given access to what is basically a partial execution trace of the program, that it can then compare to the specification.

**Strengths:**

I think this paper makes for a nice addition to the literature.
To be honest I don't think there's much in the "post-execution" part that hasn't been done already, but it's nice to see such experiments repeated with newer models and a newer dataset (LiveCodeBench), and there are some details about the experimental setup (e.g. the test synthesis) that probably haven't been done exactly the same way before.
More importantly, the discussion thereof sets up nicely for the "in-execution" experiments, and I think the authors do a great job of analysis the relative strengths and weaknesses of each approach.
Generally speaking, I am also relatively confident that the experiments were done correctly (which is more than can be said for most work in this area), and I am happy with both the model selection (one open and one API-based) as well as the datasets (I'm glad to see that the authors not only bring on board a more modern benchmark in LiveCodeBench but that they also use the less satured "Plus" versions of HumanEval and MBPP).

As a result, I think people in the code generation community would benefit from seeing this paper presented at ICLR; my concerns are relatively minor and could certainly be addressed in time for camera-ready.

**Weaknesses:**

- If I understand the method correctly, the way you are generating the trace for the "in-execution" method requires treating dynamic loops as one atomic unit. For example, if you have for loop which an unknown number of iterations, you can't statically insert a breakpoint halfway through it, so instead you have to break before or after. I would like to see this limitation discussed in the paper; do you, for example, have any thoughts on whether this hurts performance in practice, e.g. if the bug often lays within such a loop?
- I think the paper could do a better job emphasising that a +X.Y% increase in pass rate after 3 iterations of self-debugging does NOT mean that's what you want to do in practice, since it is compared against a baseline with 1 sample - not a baseline with the same budget as you used up during self-debugging. I apreciate that the point of this paper is not to chase numbers on these benchmarks but to actually do some cool science so we can understand *why* these things work the way they do, but again I am worried that a less familiar reader might mistakenly see this as a raw performance increase (neglecting the cost).
- Finally, I would encourage you to double check your references list. I was confused why you kept citing "Chen (2023)" when I was pretty sure that self-debugging paper was at ICLR 2024; indeed your references list it as ICLR 2023, which I do not believe to be correct. Probably worth double checking that you don't have any more mistakes in there.

**Questions:**

- 44-46: "Reflexion [...] evaluates [the code] with hidden oracle tests." - what do you mean by this, exactly? Is there another way of evaluating *the final* code than with the hidden oracle tests? Do you mean that they evaluate the code *before repair* with the hidden oracle tests (rather than with model-generated ones)?
- 109-110: "Self-debugging does not require increasing the sample budget, making it a cost-effective solution..." Self-debugging does of course require drawing more samples from the model, and there has been prior work showing that this is not always made up for by an increase in pass rates. I know you know this, but someone who doesn't might jump to the wrong conclusions here.
- In table 1, how can the accuracy drop from 2 to 3 iterations if you are using *oracle* tests to determine correctness? Wouldn't the passing programs from 3 iterations be a superset of those passing after 2 iterations in this case?
- 310-312: It would be nice if you could give some examples of these program contracts (maybe in an appendix). I'd be curious to see the details; in particular it's not clear whether you're just checking for type mismatches or if you have more fine-grained constraints.
- 407-409: I understand you probably don't want to open this can of worms in the paper but for the record my takeaway from your results is that data contamination is likely the reason why things look so different on LiveCodeBench in comparison to HumanEval and MBPP (which most models have at this point most certainly been trained on, knowingly or not).

---

### Official Review · Reviewer_C9Ak · 2024-11-02

**Soundness:** 3
**Presentation:** 3
**Contribution:** 3
**Rating:** 6
**Confidence:** 4

**Summary:**

This paper dives into the problem of self-debugging based on self-generated tests. It studies different programming tasks and finds that post-execution can help contest-level programming, while post-execution can enable LLMs to mitigate the bias that comes from wrong test cases by recognizing faulty feedbacks and improve performance. The experiment results demonstrate promising improvements for both basic and competitive tasks by focusing on the intermediate states.

**Strengths:**

This paper is presented very clearly and comprehensively. The motivation is reasonable: as many LLM frameworks are using self-generated tests, it is a question that whether these test cases are truly valid and accurate. And the paper answers the question well by conducting experiments on several popular simple Python programming tasks: HumanEval, MBPP, and LiveCodeBench. The results are interesting: while most of the tests are generated correctly, it is hard to guarantee that a full test suite for a question is totally correct (accuracy only 59.15% even with GPT-4o). This leads to self-debugging bias when the test cases are not fully correct.

The paper also tries different categories of execution information: in-execution and post-execution information. And they find that the latter will help offset the bias from inaccurate self-generated tests. This paper dives into the bias of post-execution information and even find that sometimes though the tests are wrong, the labels of pass or not could still help by simply adding more rounds of debugging.

These findings of the paper are interesting and could inspire research on test generation.

**Weaknesses:**

An important findings from this paper is that the in-execution state can help reduce the bias of wrong self-generated tests. However, the section of in-execution is short and not very detailed, which I think need more details:
1. A case study of how the bias is offset by in-execution information.
2. Since the in-execution helps offset the bias from wrong self-generated tests, it could be better presented in a comparison figure with post-execution information. This can show that although in-execution does not significantly improve the original performance, it does not cause harm and at least sometimes help.
3.  Though this paper provides many interesting findings, the technical side is relatively weak. It would be great if the author can provide some potential solutions on improving test generation accuracy.
4. The iteration number of the experiments is limited, which might not fully support the authors' findings.

**Questions:**

1. Can you give a case study of how the bias is offset by in-execution information?
2. Can you add the performance of more iterations of debugging and the trends?
3. Can you discuss some potential improvement on test generation?

---

### Official Review · Reviewer_qJ73 · 2024-11-03

**Soundness:** 3
**Presentation:** 3
**Contribution:** 2
**Rating:** 6
**Confidence:** 3

**Summary:**

This paper provides an investigation into the useage of self-generated tests specifically for code generation. Specifically the paper defines two types of self-debug executions: post-execution and in-execution. The paper evaluates using two different LLMs: GPT-4o and LLaMA-3-70B-Instruct across three self-contained coding benchmarks. The results show that self-debugging techniques (in particular post-execution) struggles with basic problems while in-exeuction self-debugging can effective alleviate some biases by providing the intermediate executions.

**Strengths:**

- focus an important problem of studying code generation using LLMs
- study design and settings are clear and relevant
- definitions provided in the paper provides a nice guideline for future research
- text is written clearly and easy to read

**Weaknesses:**

- **Limited number of study LLMs**:
  -  The paper aims to provide a detailed investigation to evaluate the slef-debugging capabilities of LLMs by using generated tests
  -  However, the paper only evaluates two different LLMs: GPT-4o and LLaMA-3-70B-Instruct
  -  How would the findings highlighted in the paper translate to other LLMs, for example LLMs that are not instruction-tuned or LLMs that are  specifically fine-tuned for code
  - Given the landscape of current SOTA LLMs, evaluate only on a few LLMs is not enough to showcase the investigation findings
- **Lack of more comprehensive study settings can be applied**:
  - The authors evaluate some basic settings with respect to code generation
  - Given the setup of the paper to study existing settings without providing any new technique or improvement, I would expect a more systematic settings
  - For example, the authors only use greedy sampling, what if we study the effect of sampling (for example in the classic self-debug study paper: Olausson et al. 2024) this is a key finding:
  - For example, what if we vary the number self-generated tests we produce? what if we use the tests from a larger LLM to debug the smaller LLM?
  - I believe the paper lack more interesting findings that can be obtained with more detailed experiments.
- **Evaluation on more realistic coding benchmarks**
  - The authors evaluate on some basic coding benchmarks including HumanEval, MBPP, and LiveCodeBench.
  - However, these are self-contained benchmarks that are not in fact realistic
  - I am not sure if the findings in this paper would translate to more realistic repository level coding benchmarks
  - This limitation is not neccesarily the fault of the authors as it may be difficult to perform very extensive evaluation on those benchmarks, however I would still like to see some prelimeary experiments if possible and discussion
- **Limited direction and discussion for future work**:
  - While I really like the insight of additional bias (i.e., giving more detail may not be good) in the findings and discussions, I find the discussion section after the findings rather weak.
  - I believe the paper would benefit a lot from either a small technical contribution of testing some new techniques given the findings or offer more in terms of future discussion to improve future work along this direction

**References:**
Olausson, Theo X., et al. "Is Self-Repair a Silver Bullet for Code Generation?." The Twelfth International Conference on Learning Representations. 2023.

**Questions:**

- Can the authors elaborate a bit more on how the findings in this work can lead to future improvements along this direction apart from Section 5.
- Can the authors comment on how the findings might differ when applied on more realistic benchmarks say for example for repository level coding?
- Did the authors evaluate the effect of changing the number self-generated tests and how that can affect performance?
- I am also curious how the authors deal with executions with loops in terms of basic blocks (in terms of how the intermediate states is provided to the LLM as feedback), would appreciate if more technical discussion can be provided

---

### Official Review · Reviewer_ciQm · 2024-11-04

**Soundness:** 2
**Presentation:** 3
**Contribution:** 2
**Rating:** 5
**Confidence:** 3

**Summary:**

This paper explores self-debugging in large language models (LLMs) for code generation, focusing on the use of self-generated tests. They define two paradigms in the execution-then-feedback process: post-execution and in-execution self-debugging. Experimental results across both basic and competitive code generation tasks reveal key insights. First, post-execution self-debugging struggles with simpler tasks. Second, self-generated test biases can cause inconsistencies across problem levels. Finally, in-execution self-debugging, which utilizes intermediate runtime data, consistently outperforms post-execution methods, demonstrating its promise for advancing self-debugging capabilities in LLMs

**Strengths:**

- **Analysis:** The study effectively compares two distinct self-debugging paradigms: post-execution and in-execution, providing valuable insights into their strengths and weaknesses.
- **In-Depth Analysis:** The paper delves into the limitations of post-execution self-debugging, particularly its performance degradation on simpler tasks, and offers potential explanations for this phenomenon.
- **Promising Results:** The research highlights the significant potential of in-execution self-debugging, which utilizes runtime information to enhance accuracy and reliability, especially in complex code generation scenarios

**Weaknesses:**

- **Limited Evaluation Scope:** The paper's evaluation is constrained to a relatively small set of models, potentially limiting the generalizability of the findings.
- **Novelty Concerns:** While the paper presents a novel application of self-debugging to LLMs, the core idea of leveraging runtime information for debugging is not entirely new and has been explored in previous work, as cited in [1].
- **Inconsistent Evaluation Setup:** The paper's experimental setup may introduce bias due to the different prompts used for post-execution and in-execution self-debugging. A more consistent approach would involve using the same prompts for both methods to ensure a fair comparison.

[1] Ni, Ansong, Miltiadis Allamanis, Arman Cohan, Yinlin Deng, Kensen Shi, Charles Sutton, and Pengcheng Yin. "NExT: Teaching Large Language Models to Reason about Code Execution." arXiv preprint arXiv:2404.14662 (2024).

**Questions:**

Questions

1. **Trace Length:** How will you handle extremely long traces, such as those generated by deeply nested loops? Will you implement strategies to truncate or summarize traces while preserving essential information?
2. **Consistent Evaluation:** Why is there a discrepancy in the evaluation setup for post-execution and in-execution self-debugging? For instance, let LLMs analyze input-output pairs for post-execution to decide whether it is correct and explain, which is the same as in-execution.
3. **Comparison to NExT:** How does the proposed in-execution self-debugging approach differ from NExT [1]?

---

### Author Response · Authors · 2024-12-03
**Brief summary of the rebuttal**

We sincerely thank all the reviewers for their constructive feedback, which has significantly contributed to improving our work. To provide a clear overview of our research, we would like to start by expressing our appreciation for the positive recognition of its strengths by the reviewers, including:

- The motivation is reasonable, and the presentation is clear and comprehensive (`qJ73`, `C9Ak`).
- The study effectively compares two distinct self-debugging paradigms and offers in-depth analysis (`ciQm`, `qJ73`, `C9Ak`, `edps`).
- The paper delves into the bias of post-execution information and highlights the significant potential of in-execution self-debugging (`ciQm`, `C9Ak`).
- The findings of the paper could provide valuable insights for future research (`ciQm`, `qJ73`, `C9Ak`, `edps`).

We have responded to each reviewer's questions individually and incorporated their suggestions to conduct further experiments and provide additional results and analysis in the revised version of the paper. Below is a summary of the key updates:

- We conducted additional experiments with Claude-3.5-Sonnet and Qwen2.5-Coder-7B-Instruct (Section 4).
- We included a more detailed discussion about the future work of LLMs in code generation (Section 5).
- We presented experiments with varying numbers of self-generated tests (Appendix C).
- We provided a more comprehensive case study comparing two distinct self-debugging paradigms (Section 4.2 -> Appendix A).
- We provided examples of program contracts from HumanEval and MBPP (Appendix B).
- Further proofreading and polishing were done.

---

### Meta-Review · Area_Chair_bK7e · 2024-12-19

**Metareview:**

This paper investigates having language models debug their own code by generating their own test cases, and further considers providing execution trace information to assist that debugging. It evaluates on a wide range of models, and on three (rather similar) benchmarks of function-level code generation. A relevant strength of the paper is that it does a careful study of a popular inference time technique for improving code generation, and contributes to an ongoing dialogue about the efficacy of self debugging. A primary weakness is that the benchmarks are rather narrow by 2024/2025 standards, and that it is more an evaluation of existing ideas instead of contributing novel conceptual content. Despite these positive things the paper has going for it, I recommend rejection because it does not contribute substantially novel ideas, and does not significantly expand the scope of the kinds of problems that are currently part of the ongoing empirical debate about whether self debugging works. If the paper had instead expanded the scope of methods or the scope of evaluation (eg repo-level edits), then one could see it being published.

**Additional Comments On Reviewer Discussion:**

During the discussion it was raised that relevant self debugging works had not been compared to, to which the authors revised the paper and also pointed out that their study is not quite the same in terms of its evaluation because they consider self generated tests. Other ancillary issues are eg broadening the range of language models studied, which they revised to do.

---

### Decision · Program_Chairs · 2025-01-22

Reject